# Development of a Ladder-Shape Melting Temperature Isothermal Amplification (LMTIA) Assay for the Identification of Cassava Component in Sweet Potato Starch Noodles

**DOI:** 10.3390/molecules27113414

**Published:** 2022-05-25

**Authors:** Yongqing Zhang, Yongzhen Wang, Xingmei Ouyang, Deguo Wang, Fugang Xiao, Juntao Sun

**Affiliations:** Key Laboratory of Biomarker-Based Rapid detection Technology for Food Safety of Henan Province, Xuchang University, Xuchan 461000, China; zyq336@163.com (Y.Z.); wangdg666@aliyun.com (Y.W.); zyq126@126.com (X.O.); xfug@163.com (F.X.); jtsfly@163.com (J.S.)

**Keywords:** ladder-shape melting temperature isothermal amplification (LMTIA), starch noodles, sweet potato, cassava-derived gene, authenticity

## Abstract

Food authenticity has become increasingly important as a result of food adulteration. To identify the authenticity of sweet potato starch noodles, the ladder-shape melting temperature isothermal amplification (LMTIA) method of determining cassava (*Manihot esculenta* Crantz) DNA in sweet potato starch noodles was used. A set of primers targeted at the internal transcription spacer (ITS) of cassava was designed, genomic DNA was extracted, the LMTIA reaction temperature was optimized, and the specificity of the primer was verified with the genomic DNAs of cassava, sweet potato (*Ipomoea batatas* L.), *Solanum tuberosum* L., *Zea mays* L., *Vigna radiate* L., *Triticum aestivum* L., and *Glycine max* (L.) Merr. The sensitivity with the serially diluted genomic DNA of cassava and the suitability for the DNA extracted from sweet potato starch adulterated with cassava starch were tested. The LMTIA assay for identifying the cassava component in sweet potato starch noodles was established. At the optimal temperature of 52 °C, the primers could specifically distinguish a 0.01% (*w*/*w*) cassava component added to sweet potato starch. Additionally, the LMTIA method was applied to the cassava DNA detection of 31 sweet potato starch noodle samples purchased from retail markets in China. Of these, 14 samples were positive. The LMTIA assay could be a reliable method for the rapid detection of cassava components in sweet potato starch noodles, to protect the rights of consumers and to regulate the sale market order of starch noodles.

## 1. Introduction

Sweet potato (*Ipomoea batatas* L.) containing 50–80% starch by dry matter is cultivated in over 100 countries and used as an excellent raw material for starch–based industries to produce many kinds of foods [1]. Compared with other popular starches (e.g., cassava (*Manihot esculenta* Crantz), wheat, and corn), sweet potato starch is relatively expensive [2]. Therefore, several alternative and cheaper starches have partially or completely replaced sweet potato starch in the production of sweet potato starch products. For example, sweet potato starch noodles, which have the characteristics of unique noodle texture and excellent transparency, are produced from sweet potato starch and are primarily consumed in Asian countries [2]. A study that detected the cassava component in sweet potato noodles showed that 57.7% of sweet potato noodle products (30/52) from retail markets in China were adulterated with cassava starch [3]. This type of adulteration seriously harms the rights of consumers and disturbs the sale market order. Therefore, the determination of the starch origin in sweet potato starch products, especially the determination of a cassava component in sweet potato starch noodles, is required, in order to facilitate surveillance of commercial adulteration.

For starch origin investigations, some studies have used various approaches or techniques such as infrared spectroscopic techniques [4,5,6], peptide mass fingerprinting [7], Fourier transform infrared spectroscopy [8], chromatography [9], and separation settling methods [10]. These methods identify different starches by the properties (e.g., the morphology and structure) of the starch granules, but they have some limitations, namely, the use of expensive instruments and difficulties in achieving sampling uniformity and sampling representativeness. Methods based on DNA, such as polymerase chain reaction (PCR) and DNA barcoding methods, are very effective for the determination of the authenticity of starch products [11,12,13]. Moreover, droplet digital PCR [14] and real–time PCR [15] methods have also been developed for the detection of the cassava component in edible starch. DNA–based methods rely on expensive equipment and require skilled personnel. Developing a rapid, sensitive, and cost–effective method for the determination of adulterated starch is a pressing matter.

Loop–mediated isothermal amplification (LAMP) is used to amplify nucleic acids under isothermal conditions [16]. This method has the properties of rapidity, sensitivity, and specificity and can be applied in various fields [17,18]. It can be used to detect GMO maize starch [19] and to discriminate the cassava component in sweet potato noodles [3].

Based on LAMP, the novel method of ladder-shape melting temperature isothermal amplification (LMTIA) was developed in 2021 [20]. The LMTIA technique was used with one pair of primers or two pairs of nested primers and a thermostable DNA polymerase to amplify the target sequence on the internal transcribed spacer of *Oryza sativa*, with a ladder-shape melting temperature curve. This simple and rapid method was developed for the authentication and determination of plant–derived foods. Compared with LAMP, LMTIA yields robust amplification of nucleic acids with high specificity and sensitivity in very short time [20].

The objective of this study was to develop an LMTIA method for rapid and specific determination of cassava DNA when cassava starch is used in sweet potato starch noodles. The method developed needed to be rapid and easy enough for routine use. Our results are useful for developing a detection technology for the authentication and adulteration analysis of foods.

## 2. Results

### 2.1. Optimization of the LMTIA Reaction Temperature

As shown in Figure 1, for LMTIA reactions at 51 °C, 52 °C, 53 °C, and 54 °C, the results of all positive controls using the cassava DNA as a template were positive. The results of all negative controls, in which the cassava DNA template was substituted by ddH_2_O, were negative. When the temperature was 52 °C, the best repeatability and the highest efficiency were indicated by the amplification plot and melt curve. Additionally, The LMTIA reaction entered into the exponential amplification stage when the number of cycles was 14. Therefore, the optimal temperature was 52 °C, and the LMTIA assay was rapid.

### 2.2. Specificity Determination of the LMTIA Assay

The specificities of the LMTIA assay were determined at 52 °C with the genomic DNA of cassava, sweet potato, *S. tuberosum*, *Z. mays*, *V. radiata*, *T. aestivum*, and *G. max*. As shown in Figure 2, the LMTIA reactions with the cassava DNA as a template had exponential amplification, whereas the other reactions had no exponential amplification. Therefore, the LMTIA assay was of high specificity and only amplified the cassava DNA.

### 2.3. Sensitivity Determination of the LMTIA Assay

The sensitivities of the LMTIA assay were determined at 52 °C with amounts of cassava DNA ranging from 1 ng to 0.1 pg. As shown in Figure 3, the LMTIA reactions using the cassava DNA in amounts of 1 ng, 100 pg, 10 pg, and 1 pg as templates had exponential amplification, whereas the reactions with the cassava DNA at 0.1 pg and the negative controls, in which the cassava DNA template was substituted by ddH_2_O, had no exponential amplification. Therefore, the sensitivity of the LMTIA assay corresponded to 1 pg of cassava DNA.

### 2.4. Suitability Determination of the LMTIA Assay

The suitability of the LMTIA assay was determined at 52 °C with the cassava DNA extracted from the sweet potato starch, which was adulterated with cassava starch at 5%, 1%, 0.1%, and 0.01% (*w*/*w*). As shown in Figure 4, the LMTIA reactions using the extracted cassava DNA as a template had exponential amplification, whereas the negative controls, in which the DNA template was substituted by ddH_2_O, had no exponential amplification. Therefore, the detection limit of the LMTIA assay for the cassava–adulterated sweet potato starch was 0.01%.

### 2.5. Cassava Component Detection in Sweet Potato Starch Noodles

A total of 31 samples labeled as sweet potato starch noodles were collected from retail markets in China. Of these, 14 samples tested positive for cassava DNA, according to the results of the LMTIA assay, indicating that cassava starch is typically used to process sweet potato starch noodles in China but without clear identification.

## 3. Discussion

As reported in the previous paper [20], the LMTIA technique based on LAMP can be used under isothermal conditions to detect *Oryza sativa* L. DNA with 50 times more sensitivity than LAMP. The nucleic acid sequence with a ladder–type melting temperature curve on the internal transcribed spacer (ITS) of cassava was selected using the Oligo 7 software (Molecular Biology Insights, Inc., Cascade, CO, USA) and aligned in GenBank. Based on the selected sequence, the LMTIA primers were designed using the online software Primer3Plus (https://dev.primer3plus.com/index.html, accessed on 28 July 2021), and they displayed high specificity at 52 °C for the cassava DNA (Figure 1 and Figure 2). This simple and rapid method could be applied to determine the authenticity of plant–derived food.

In recent years, food fraud has become an urgent problem linked to the traceability of food products throughout the supply chain from crop to supermarkets [21]. Food authenticity has become increasingly important as a result of food adulteration. Development of selective and sensitive detection techniques is a key challenge for the development of authentic food, and this limitation prevents the widespread use of cheap food sources in the food industry. In the field of starch products, researchers have explored determination methods for the origin of starch, which is very important for identifying the authenticity and adulteration of starch–based foods. Cho et al. (2013) [7] reported that the peptide mass fingerprinting technique could successfully identify the origin of the starches (with higher than 10% adulteration) contained in starch noodles. Xu et al. (2013) [6] reported that near–infrared spectroscopy could distinguish a 5% adulteration with sweet potato, cassava, potato, or corn components in lotus root powder. As described in a previous paper [3], the real–time LAMP method could accurately and specifically detect the cassava component in sweet potato noodles, with a detection limit of 1%. In this study, the sensitivity of the LMTIA assay for 10–fold dilutions of cassava DNA (1 ng, 100 pg, 10 pg, 1 pg, and 0.1 pg) was 1 pg (Figure 3), and the primers could also specifically distinguish a 0.01% (*w*/*w*) cassava component added to sweet potato starch (Figure 4). Therefore, the developed LMTIA method had high sensitivity and could be applied to the monitoring of the adulteration of sweet potato starch noodles with cassava starch. In addition, due to the low requirement for equipment and skilled personnel, LMTIA assays can be used to detect other food frauds and pathogens, possibly including SARS-CoV-2. Continuous technological developments related to raw materials processing, quality enhancement, and new determination methods contribute to the growing popularity and acceptability of cheap food sources. The newly developed LMTIA assay could lead to the development of cassava starch noodles as an alternative to sweet potato starch noodles, as well as alternatives to other expensive starch products.

## 4. Materials and Methods

### 4.1. Selection of the Target Sequences for LMTIA Primer Design

There are three criteria for the target sequence selection, as described by Wang et al. [20]: the melting temperature curve of the sequence is of a ladder type; the GC content of the sequence is generally 40–80%; and the sequence has high specificity. According to the published gene sequence of the cassava internal transcribed spacer (ITS) in GenBank (accession number: MK114629.1), the nucleic acid sequence was analyzed using Oligo 7 software (Molecular Biology Insights, Inc., Cascade, CO, USA) and aligned in GenBank. The melting temperature curve of the 66 nt sequence fragment was of a ladder type, as shown in Figure 5. Its GC content was 68.13%, and the sequence was highly specific to species of *M. esculenta*. Therefore, the 66 nt sequence was selected as the target sequence for the LMTIA primer design.

### 4.2. Primer Design for LMTIA Assay

The primers of the LMTIA were designed with the above selected target sequence by the online software Primer3Plus (https://dev.primer3plus.com/index.html, accessed on 28 July 2021). According to the report [20], the Tm range for the outer pair of primers P1 and D1 was between 59 °C and 63 °C, and the length range was between 18 and 25 nt. The Tm range for the inner pair of primers P2 and D2 was between 35 °C and 85 °C, and the length range was between 12 and 16 nt. As shown in Table 1, the primer P was from the reverse complementary sequence of primer P2 and primer P1 linked with –tttt–, and the primer D was from the reverse complementary sequence of primer D2 and primer D1 linked with –tttt– [20]. The primers P and D were synthetized by General Biosystems (Anhui) Co., Ltd.

### 4.3. Genomic DNA Extraction

Genomic DNA samples were extracted from cassava, sweet potato, *S. tuberosum*, *Z. mays*, *V. radiata*, *T. aestivum*, and *G. max* using a NucleoSpin® Food kit (Macherey–Nagel GmbH & Co. KG, Düren, Germany) according to the manufacturer’s instructions.

### 4.4. Optimization of the LMTIA Reaction Temperature

The LMTIA assay that used the above synthetized primers was carried out in a 10 μL reaction mixture, as described by Wang et al. [20], with 0.8 µM of each primer (primers P and D), 1.0 mM dNTPs, 1× reaction buffer (20 mM Tris–HCl (pH 8.8), 10 mM KCl, 10 mM (NH_4_)_2_SO_4_, 6 mM MgSO_4_, 0.1% Triton X–100), 3.2 U Bst DNA polymerase (Merit Biotech [Shandong] Co., Ltd., Heze, China), 1× SYBR Green Ⅰ, and 1 µL of cassava DNA. Liquid paraffin was added (20 µL) to avoid aerosol contamination after the preparation of the reaction mixture. The reaction mixture was heated at 51 °C, 52 °C, 53 °C, and 54 °C for 60 min (90 s per cycle just for gathering fluorescence signals every 90 s). The amplification plot and melt curve were obtained using a StepOne^TM^ System (Applied Biosystems, Foster City, CA, USA).

### 4.5. Specificity Determination of the LMTIA Assay

The DNAs of cassava, *I. batatas*, *S. tuberosum*, *Z. mays*, *V. radiata*, *T. aestivum*, and *G. max* were used to determine the specificity of the LMTIA assay. The amount of the DNA template used was 1 µL per reaction, and the reaction mixture was heated at the optimized temperature of 52 °C for 60 min.

### 4.6. Sensitivity Determination of the LMTIA Assay

The sensitivity of the LMTIA assay was detected with 10–fold dilutions of genomic DNA from cassava (1 ng, 100 pg, 10 pg, 1 pg, and 0.1 pg), and the reaction mixture was heated at the optimized temperature of 52 °C for 60 min.

### 4.7. Suitability Determination of the LMTIA Assay

The sweet potato starch was blended with cassava starch at 5%, 1%, 0.1%, and 0.01% (*w*/*w*). The DNAs were isolated using the NucleoSpin Food kit (Macherey–Nagel GmbH & Co. KG, Düren, Germany) according to the manufacturer’s instructions. The detection limit was determined using the LMTIA method at the optimized temperature of 52 °C for 60 min.

### 4.8. Cassava Component Detection in Sweet Potato Starch Noodles

A total of 31 samples labeled as sweet potato starch noodles were obtained from retail markets in China. The DNAs of the samples were extracted using the NucleoSpin Food kit (Macherey–Nagel GmbH & Co. KG, Düren, Germany) according to the manufacturer’s instructions. The cassava component in the samples was tested using the LMTIA method at the optimized temperature of 52 °C for 60 min.

## 5. Conclusions

To detect cassava DNA, the LMTIA primers were designed with a 66 nt sequence of cassava ITS gene (GenBank accession number: MK114629.1) as the target. The melting temperature curve was of a ladder type, and an LMTIA assay for the identification of the cassava component in sweet potato starch noodles was developed. At the optimal temperature of 52 °C, the LMTIA primer could specifically distinguish the cassava DNA from the DNAs of sweet potato, *S. tuberosum*, *Z. mays*, *V. radiata*, *T. aestivum*, and *G. max* and could also identify a 0.01% (*w*/*w*) cassava component added to sweet potato starch. The effectiveness of the method was further verified by the detection of a cassava component in 31 samples labeled as sweet potato starch noodles and sold on the market. Therefore, the rapid and simple LMTIA assay has promise for identifying food authenticity, facilitating surveillance of commercial adulteration, and protecting consumer rights.

## Figures and Tables

**Figure 1 molecules-27-03414-f001:**
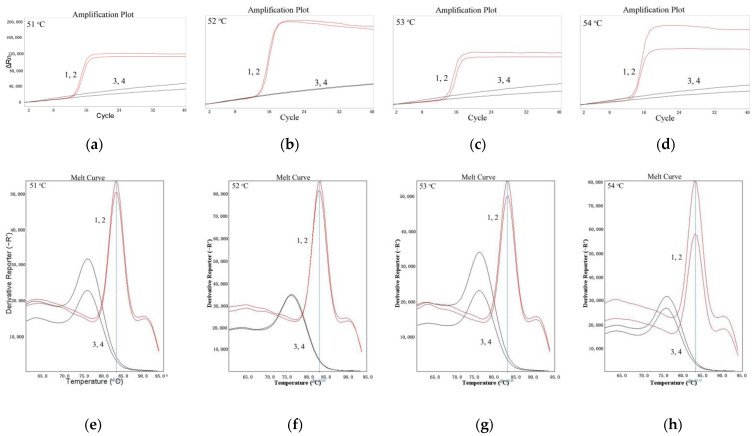
Amplification plot and melt curve of the LMTIA reaction at different temperatures. (**a**): the amplification plot of the LMTIA reaction at 51 °C; (**b**): the amplification plot of the LMTIA reaction at 52 °C; (**c**) the amplification plot of the LMTIA reaction at 53 °C; (**d**): the amplification plot of the LMTIA reaction at 54 °C; (**e**): the melt curve of the LMTIA reaction at 51 °C; (**f**): the melt curve of the LMTIA reaction at 52 °C; (**g**): the melt curve of the LMTIA reaction at 53 °C; (**h**): the melt curve of the LMTIA reaction at 54 °C. 1, 2: cassava (*M. esculenta*) DNA; 3, 4: ddH_2_O.

**Figure 2 molecules-27-03414-f002:**
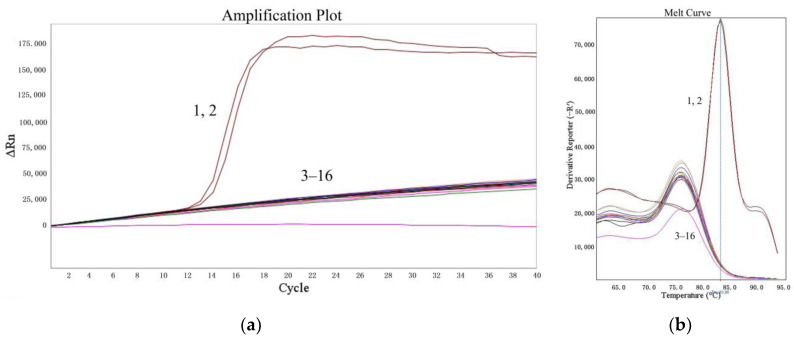
Specificity of the LMTIA assay with different DNA types. (**a**): the amplification plot of the LMTIA assay; (**b**): the melt curve of the LMTIA assay. 1, 2: cassava DNA; 3, 4: ddH_2_O; 5–16: the DNAs of *I. batatas*, *S. tuberosum*, *Z. mays*, *V. radiata*, *T. aestivum*, and *G. max*.

**Figure 3 molecules-27-03414-f003:**
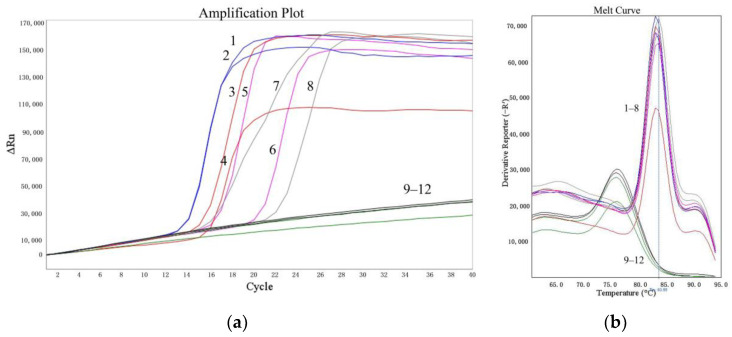
Sensitivity of the LMTIA assay with serial dilution of cassava DNA. (**a**): the amplification plot of the LMTIA assay; (**b**): the melt curve of the LMTIA assay 1, 2: 1 ng; 3, 4: 100 pg; 5, 6: 10 pg; 7, 8: 1 pg; 9, 10: 0.1 pg; 11, 12: ddH_2_O.

**Figure 4 molecules-27-03414-f004:**
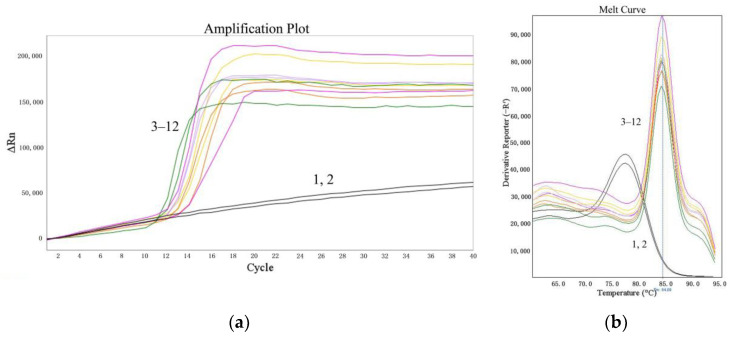
Suitability of the LMTIA assay for distinguishing the DNA extracted from sweet potato starch mixed with cassava starch. (**a**): the amplification plot of the LMTIA assay; (**b**): the melt curve of the LMTIA assay. 1, 2: ddH_2_O; 3, 4: 1 ng cassava DNA; 5–12: the DNA extracted from sweet potato starch adulterated with cassava starch at 5%, 1%, 0.1%, and 0.01% (*w*/*w*).

**Figure 5 molecules-27-03414-f005:**
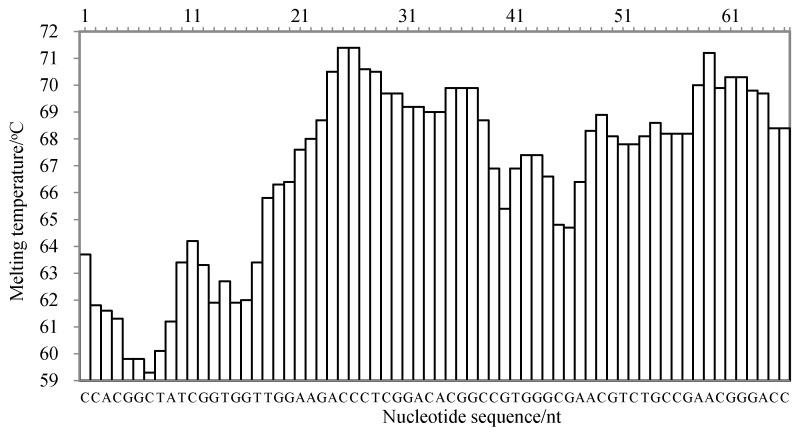
The melting temperature curve of the target sequence for the detection of the internal transcribed spacer of cassava using the LMTIA method.

**Table 1 molecules-27-03414-t001:** LMTIA primers for the internal transcribed spacer of cassava.

Primer	Sequence (5′–3′)
P (P2–tttt–P1)	TGTCCGAGGGTCTTTTTTCACGGCTATCGGTGGT
D (D2–tttt–D1)	GCCGTGGGCGAACTTTTGGTCCCGTTCGGCAGAC

## Data Availability

The data presented in this study are available on request from the corresponding author.

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
