# Peer review of "Development of a Ladder-Shape Melting Temperature Isothermal Amplification (LMTIA) Assay for the Identification of Cassava Component in Sweet Potato Starch Noodles"

_molecules, 2022, doi:10.3390/molecules27113414_

Round 1

Reviewer 1 Report

The current manuscript reports the development of a ladder-shape melting temperature isothermal amplification (LMTIA) assay for the identification of cassava component in sweet potato starch noodles.

In general, this is an important and interesting work. I have, however a few comments or suggestions.

  • The conclusions regarding the optimal parameters are not convincing, due to the fact that these turned out to be borderline values. In that case, why didn't they do additional studies, for example, LMTIA reactions at 51 ° C, and at a lower concentration of cassava starch when diluted?
  • The classical structure of a scientific article is violated, the Results and Discussion are in front of Materials and Methods, there are no formulated Conclusions
  • Part of the text in a lighter color

Reviewer 2 Report

Food adulteration has been a problem in the food industry for centuries, that why food authenticity has become increasingly important. Therefore, I believe that the strength of the article is to undertake research on this topic. However, the article contains some shortcomings that should be corrected.

Abstract

  1. 14 Manihot aesculenta Crantz. Correct and add
  2. 18 Ipomoea batatas L. Solanum tuberosum L., Zea mays L., Vigna radiata L., Triticum aestivum L., and Glycine max (L.) Merr. add

Introduction

l.32 (Ipomoea batatas L.) add

l.35 (Manihot esculenta Crantz.) add

l.73 Transfer the entire two sentences to results     The developed method needs to be rapid and easy enough for routine use. Our 73 results are useful for developing a detection technology for the authentication and adul-74 teration of food.

Results

  1. 86 convert to S.tuberosum, Z. mays, V. radiata, T. aestivum and G. max

l.96 convert to M. aesculenta

l 99 correct as above

Discussion

  1. 131 add Oryza sativa L.

Materials and methods

why the chapter material and methods does not appear at the beginning of the article, after the introduction

l.193 convert S. tuberosum, Z. mays, V. radiata, T. aestivum and G. max

l.209 correct as above

subsection 4.8 should be at the beginning of Materials and methods

please write conclusions that are well worded in the abstract

Round 2

Reviewer 1 Report

Text edits accepted